# GUIDE YOUR ANOMALY WITH LANGUAGE

## ABSTRACT

Anomaly detection is the task of identifying data that is different from what is considered normal. Recent advances in deep learning have improved the performance of anomaly detection and are used in many applications. However, it can be difficult to create a model that reflects the desired normality due to various issues, including lack of data and nuisance factors. To address this, there have been studies that provide the desired knowledge to the model in various ways, but there are limitations, such as the need to understand deep learning. In this work, we propose a method to guide the desired normality boundary in an image anomaly detection task using natural language. By leveraging the robust generalization capabilities of the vision-language model, we present **L**anguage-**A**ssisted **F**eature **T**ransformation. LAFT transforms image features to suit the task through natural language using the shared image-text embedding space of CLIP. We extensively analyze the effectiveness of the concept on a toy dataset and show that it works effectively on real-world datasets.

## 1 INTRODUCTION

### 1.1 MODELING NORMALITY FOR ANOMALY DETECTION

Anomaly detection is the task of distinguishing abnormal data that are different from normal data. With the recent development of deep learning, the performance of anomaly detection has improved considerably and is widely used in applications such as industrial anomaly detection and video anomaly detection. To detect abnormalities effectively, deep learning models should be able to learn the concept of normality. Typically, the user provides the model with normal samples to learn from. However, it can be challenging to obtain all the possible variations of the samples and to differentiate anomalies due to nuisance factors in the data (Cohen et al., 2022).

In practical applications, there are cases where the model should pay attention to or disregard certain attributes. Here are some motivating examples. (1) When inspecting a product from an image, users may only be interested in the shape of the product, not its position, angle, or setting in which it was taken. In this situation, the model should focus solely on the shape of the product. (2) When performing anomaly detection in CCTV, the change in brightness is irrelevant and only the content such as the movement of the object is important. (3) There are also situations where it is difficult to distinguish anomalies due to entangled attributes. For example, the background and birds are entangled in the Waterbirds dataset (Sagawa et al., 2019).

In order to address this issue, there have been attempts to generate additional data through data augmentation or data generation to better learn the decision boundary (Zavrtanik et al., 2021; Li et al., 2021; Du et al., 2021). The aim of these methods is to create samples more diverse than what is available, so that the model can more accurately distinguish between normal and abnormal data. However, these methods should be able to generate the desired normality boundary by adding the characteristics of outliers through appropriate augmentation or generation techniques. Furthermore, there have been attempts to make models learn task-specific feature representations (Chen et al., 2020a;b; Caron et al., 2020) and apply them to anomaly detection to better learn normality at the feature level (Hyun et al., 2023). To make use of pre-trained backbones that are trained task-agnostic, there have been studies that involve fine-tuning the feature extraction backbone or creating task-specific features through feature transformation (Caron et al., 2020; Reiss & Hoshen, 2023; Tack et al., 2020). However, the downside is that it is costly to fine-tune the backbone or train the transformation, and it is difficult to properly learn the desired anomaly at the feature level.

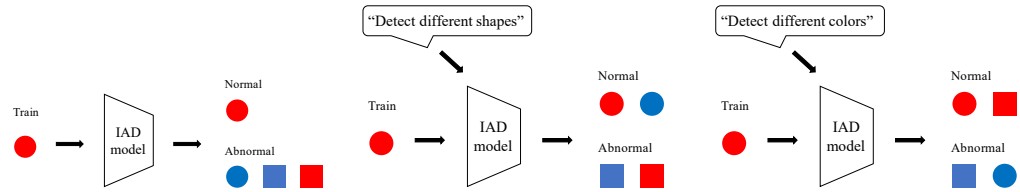

Figure 1: High level overview of our goal. We want users to be able to give the model a sense of normality through language.

## 1.2 VISION-LANGUAGE MODELS IN ANOMALY DETECTION

Research in the field of natural language processing has shown the effectiveness of training models with extensive, unlabeled Internet data, and this approach has also been applied to computer vision (Radford et al., 2021; Jia et al., 2021; Desai et al., 2023). They demonstrated the effectiveness of using image-text pairs obtained from the Internet to pre-train models, integrating natural language description to enhance the quality of image representations. Models trained at scale in this manner can establish connections between visual concepts in images and natural language descriptions, aligning image and text features within a shared embedding space. These models can extract remarkably general representations and show impressive performance in downstream tasks.

Many researchers are trying to use the powerful performance of vision-language models in the field of image anomaly detection. In particular, there are studies that apply vision-language models to industrial anomaly detection (Jeong et al., 2023; Cao et al., 2023; Chen et al., 2023), or general image out-of-detection tasks by utilizing the characteristics that vision-language models can be applied to downstream tasks with zero-shot using text prompts (Ming et al., 2022; Miyai et al., 2023). The advantage is that human prior knowledge can be fed to the model using text prompts, allowing for zero-shot use without training images. Models that take this approach usually define normality using text prompts and calculate anomaly scores using the similarity of text and image features. However, in some cases, it is difficult to define normality using natural language alone, and it is common to use reference image features in conjunction with text features to define normality. Comparison of the reference image and the target image is done at the feature level, which means that even a slight difference between the two images can cause a drastic change in similarity and reduce scalability.

## 1.3 GUIDE YOUR ANOMALY WITH LANGUAGE

As discussed in subsection 1.1, it is difficult to define the boundary of in-distribution using only images, and a thorough understanding of deep learning is also required. This includes the selection of expertly designed image transformations to reflect user knowledge. In subsection 1.2, we discussed the difficulty in defining normality using only language or a few reference images. In this paper, we propose a method that enables users to define the boundaries of normality for images using language, taking advantage of the properties of CLIP (Radford et al., 2021). Our approach is different from the majority of existing work, as it relies mainly on image features to define normality, with language playing a supporting role. By using language, users can "guide" normality, giving them more flexibility to incorporate their knowledge of what is normal. Additionally, by setting the boundaries of normality with the image features, we can accurately distinguish between normal and abnormal images.

We summarize our contributions as follows:

1. We propose Language Assisted Feature Transformation (LAFT), a method that uses natural language to transform image features to suit the task at hand. This is achieved by taking advantage of the strong generalization capabilities of a vision-language model and an image-text aligned embedding space.

2. We introduce LAFT AD, a method for anomaly detection that can focus or ignore image attributes in natural language, using LAFT.

3. We extensively examine the performance of our method on a simple dataset and demonstrate that it is successful on real-world datasets.

## 2 RELATED WORK

**Image anomaly detection with vision-language model**    Starting with Radford et al. (2021) as a basic vision language model, the field of image anomaly detection has seen remarkable progress. Ming et al. (2022) introduced a novel scoring method, which was refined in the updated version, (Miyai et al., 2023), to improve the accuracy of anomaly detection. To address the challenge of out-of-distribution detection, Ming & Li (2023) proposed a parameter-efficient training approach, highlighting the nuances of fine-tuning for this task. In Fort et al. (2021), a method of feeding potential out-of-distribution labels to the CLIP text encoder was introduced. In addition, Esmaeilpour et al. (2022) presented a strategy for training a label generator based on the CLIP image encoder for out-of-distribution detection, although it focused primarily on small inputs.

**Anomaly detection with type control**    In addressing anomaly detection with type control problem, several methodologies have been explored. Wang et al. (2022) delves into disentangling the factors of variation in the data. The with-language methodology, as illustrated in El Banani et al. (2023) employs contrastive representation learning guided by a vision-language model, improving feature learning for anomaly detection. Cohen et al. (2022) advocates for labeling all attributes, providing a structured framework to potentially improve the robustness of anomaly detection models. However, Reiss et al. (2023) underscores an inherent limitation, emphasizing that no single method can be universally applied to all anomaly detection problems, thus necessitating a nuanced, problem-specific methodology in this domain.

**Feature adaptation**    In the context of feature transformation for anomaly detection, several strategies have been developed to enhance the adaptability and robustness of backbone models. Ruff et al. (2018) have a different strategy, starting with the pre-training of a representation encoder through autoencoding on regular data, creating a basis for subsequent anomaly detection activities. Chen et al. (2020a;b) effectively employs contrastive pre-training to facilitate feature agreement, particularly advantageous for downstream anomaly detection. Caron et al. (2020) utilizes prototype vectors for contrastive training of similar features, leading to the refinement of feature representations. Subsequently, these approaches are adapted for One-Class Classification (OCC) objectives using techniques such as those proposed by Reiss et al. (2021); Hyun et al. (2023); Reiss & Hoshen (2023). However, the adaptation process often faces challenges, including the issue of catastrophic collapse.

## 3 PRELIMINARIES

In our scenario, the training set, represented as $\mathcal{D}_{\text{train}}$, comprises solely of normal samples. We then define the normality within the image features. Our evaluation set $\mathcal{D}_{\text{test}}$ consists of normal and anomalous samples. The attribute labels ($0 \leq j < m$) for a test image $x_i$ are denoted as $y_i = (y_0, \cdots, y_{m-1})_i$. The $m$ attributes can be divided into relevant ($0 \leq j < n$) and irrelevant ($n \leq j < m$) categories, with examples such as the object's identity, color saturation, and background noise representing different attributes. We assume that $n$ is not a fixed number but uncertain and that the anomaly label is always a function of (potentially) all relevant attributes $y_i = f^a(y_0, \cdots, y_{n-1})$. That is, the nuisance attribute $y_n, \cdots, y_{m-1}$ never affects the anomaly label $y_i$. We emphasize that in our described setting, neither the relevant attribute labels nor the anomaly labels are given.

Our goal is to transform the feature vector $T(f_i) = T(f(x_i)) \in \mathbb{R}^d$ using the transformation function $T$ into a target space that significantly distinguishes between normal and abnormal data, implying that $T$ distills the information of irrelevant features. That is, we desire our transformation function to represent the relevant attributes in a manner unaffected by the nuisance attributes:

$$p(y_n, \cdots, y_{m-1}) = p(y_n, \cdots, y_{m-1}|T(f_i)). \tag{1}$$

We also wish our code to be informative - to represent sufficient information regarding our relevant attributes ($I(;)$ is the mutual information between its two arguments):

$$I((y_0, \cdots, y_{n-1}); f_i) \sim I((y_0, \cdots, y_{n-1}); T(f_i)). \tag{2}$$

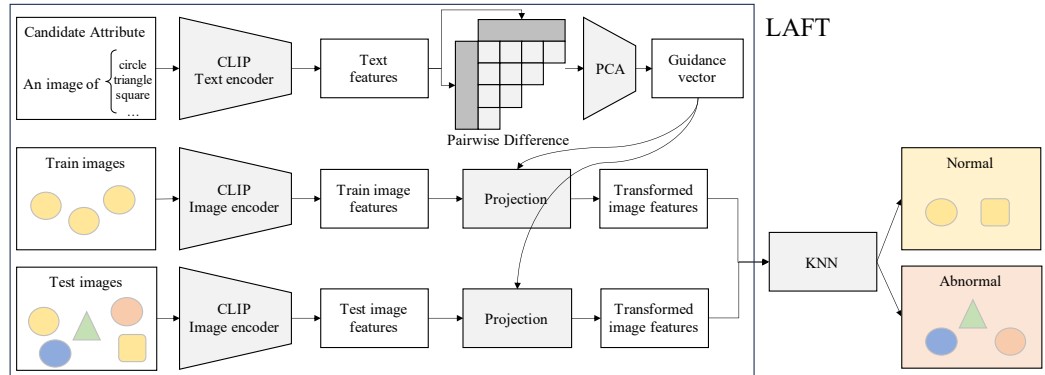

Figure 2: Overview of our method.

In practice, invariance can be measured by the accuracy of predicting $y_i = (y_n, \cdots, y_{m-1})_i$ from the transformed code $T(f_i)$. But, we can assess the informativeness by measuring the accuracy of predicting the relevant attribute utilized to define anomalies. Empirical evaluations of these measures for our datasets can be found in the next section. With such a representation, we may later evaluate anomalies independently, devoid of any bias caused by the irrelevant attribute we aim to disregard.

CLIP (Radford et al., 2021) embeds the features in a unit sphere subspace in Euclidean space $\mathbb{R}^n$. An embedding vector of an image is correlated to the text embedding describing the image. This means that we can construct the transform with the CLIP text encoder $T_{text}$. We assume that all relevant and irrelevant features can be encoded with the text description, so that natural language assists the manipulation of the vector in the CLIP space:

$$I((y_0, \cdots, y_{n-1}); f_i) \sim I((y_0, \cdots, y_{n-1}); T_{text}(f_i)). \qquad (3)$$

## 4 METHOD

Our main goal is to transform visual features with text guidance without any further training. Typical learning-based methods need to collect data to get a feasible normality space and require many computing powers to train deep neural networks. In this section, we describe a way to distill undesirable attributes with vector projection with the help of provided text prompts.

### 4.1 TEXT PROMPT

To enable the model to focus on or ignore certain attributes of the image, it is necessary to provide the model with proper textual prompts. Similar to (Ming et al., 2022), we assume that the text contains the "concept prototype" for the attributes. So we give the model a list of prompts, each prompt consisting of the following form: TEMPLATE + ATTRIBUTE_VALUE. For example, if we want to ignore the color of the hair, we can construct the prompt as follows:

- "a photo of a person with brown hair"
- "a photo of a person with black hair"
- "a photo of a person with blond hair"
- "a photo of a person with gray hair"
- . . .

Using the actual values of the desired attribute in the prompts, we want the model to know the difference between the visual concepts associated with the attribute. Providing values corresponding to this attribute that are not actually in the training set but are likely to appear at test time makes it easier to construct a subspace for that concept. This would be prior user knowledge. As with other language-based methods, you can use multiple types of template to mitigate the bias introduced by the template itself.

### 4.2 FIND CONCEPT SUBSPACE

After the user gives a prompt, our method tries to find the subspace of visual concepts in which the attribute exists from the given prompt. Specifically, we find the axis along which the variance

between concepts is represented by the concept difference between prompts. For prompts $t_i$ and $t_j$ where $1 \leq i < j \leq n$, we compute the pairwise difference:

$$v_{ij} = E_{text}(t_i) - E_{text}(t_j)$$

where $n$ is the number of prompts and $E_{text}$ is the CLIP's text encoder. We call these vectors "concept axis". But directly using these vectors as basis is not preferable because the text prompt itself may contain irrelevant information about the target attribute. So, we extract the principle axis from these vectors using PCA:

$$\{g_k\} = \text{PCA}(\{v_{ij}\}, d)$$

where $d$ is the number of components and $\{g_k\}$ is the $d$ principal axes and named a set of **guidance vectors**. Throughout the paper, we typically choose $d$ from 4 to 32 when guiding an attribute and from 32 to 384 when ignoring an attribute. We construct the attribute subspace using these principal axes as basis vectors.

### 4.3    FEATURE TRANSFORMATION WITH PROJECTION

For all image feature vectors $f_i = f(x_i)$ encoded by the CLIP image encoder, we project the features with the guidance vector $g_k$:

$$\hat{f}_i = <f_i, g_k> g_k, \tag{4}$$

where the notation $< \cdot, \cdot >$ is the inner product. This projection cancels out the other direction in the context of suppressing irrelevant attributes. Without loss of generality, we can further enlarge the number of relevant attributes to two or more. Then from the attributes, we can generate the guidance vectors $g_k$. Then we can project on the space generated by the $g_k$'s or write it:

$$\hat{f}_i = \sum_{k=1}^{m} <f_i, g_k> g_k, \tag{5}$$

where $m$ is the number of guidance vectors.

On the other hand, we can also 'relaxation' to the irrelevant attributes using orthogonal projection. Let $\bar{g}_k$ be the guidance vectors generated by the irrelevant attributes. Then we can orthogonally project onto the space generated by $\bar{g}_k$'s:

$$\hat{f}_i = f_i - \sum_{k=1}^{\bar{m}} <f_i, \bar{g}_k> \bar{g}_k. \tag{6}$$

Contrary to the inner project, we can manually cancel out the vectors of irrelevant attributes. By doing so, the specification of the normality can be gained in our anomaly detection task.

### 4.4    DENSITY BASED ANOMALY SCORING

We operate under the assumption that the mapping will place anomalous samples in areas of sparse concentration, while normal data will be allocated to areas of dense concentration, resembling the behavior observed in other anomaly detection methodologies. In a scenario where the representation is exclusively composed of relevant attributes, it's likely that regions exhibiting low density would correspond to samples characterized by uncommon relevant attributes, which are likely to be classified as anomalies. To numerically estimate the density of the normal data around each test sample, we use the $k$-nearest-neighbors algorithm ($k$NN). We begin by extracting the representation for each normal sample: $f_i^t = T(f(x_i))$, $\forall x_i \in \mathcal{D}_{train}$. Next, for each test sample, we infer its latent $f_{test}^t = T(f(x_{test}))$. Finally, we score it by the distance $k$ NN from the normal data:

$$S(x_t) = \frac{1}{k} \sum_{f_i^t \in N_k(f_t^t)} sim(f_i^t, f_t^t) \tag{7}$$

where $N_k(f_t^t)$ denotes the $k$ most similar relevant attribute transformed feature vectors in the normal data. We use $k = 30$ throughout paper for $k$NN. We note that the high dimension of the latent space allows us to distinguish between high and low-density areas of the distribution of normal data.

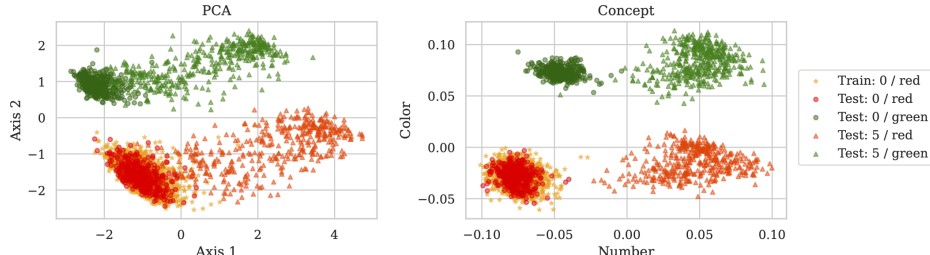

Figure 3: The features of the train and test images are mapped linearly onto two sets of axes: (**left**) the PCA axes and (**right**) the concept axes. We calculate both axes using image features from the auxiliary train set (not plotted) and reduce attribute values for visualization.

## 5 EXPERIMENTS

### 5.1 SETUP

**Models and Prompts** Throughout the paper, we use the CLIP ViT-B/16 model with the pre-trained checkpoint from OpenAI [1]. For a fair comparison, we also adopted the CLIP ViT-B/16 image encoder as a feature extractor for the baseline methods. We also use the same text prompts for the methods using the CLIP text encoder and our method. See Appendix A for details of the prompts we used in the experiments.

**Datasets** To validate our approach, we employ the colored version of the MNIST (LeCun et al., 2010), Waterbirds (Sagawa et al., 2019), and CelebA (Liu et al., 2015) datasets. We set the normal and abnormal values for each attribute of the dataset and divide the train split into $2^m$ subsets. For instance, in the Colored MNIST dataset, we designate the numbers 0-4 as normal and 5-9 as anomalous, and the color red as normal and the color green and blue as anomalous. We then use one subset as a train set that is considered normal in all $m$ attributes (e.g. 0-4 and red).

We consider two scenarios: a standard image anomaly detection task that does not have access to abnormal or external samples, and a more relaxed setting that can use a few abnormal or external samples. In other words, in the latter situation, the model can use a few samples from a different subset in the training set (e.g. 0-4 and blue). This setting is similar to outlier exposure (Liznerski et al., 2022). When using the auxiliary dataset of a few shots, we randomly sample the data of $k$ in each subset.

**Baselines** We use $k$NN for our method and as a baseline. We directly use image features from the CLIP image encoder to compute $k$NN distances. For our method, LAFT AD, we transform the image features as described in section 4 before $k$NN. Mean-Shifted Anomaly Detection (**MSAD**; Reiss & Hoshen, 2023) transforms the pre-trained representations to better fit the anomaly detection in an unsupervised manner. We also consider CLIP-based anomaly detection methods. Maximum Concept Matching (**MCM**; Ming et al., 2022) is a CLIP based out-of-distribution detection method. This only requires prompts for normal images for anomaly scoring. On the other hand, the method proposed by Fort et al. (2021) and Zero-shot OOD detection based on CLIP (**ZOC**; Esmaeilpour et al., 2022) uses candidate prompts for anomalous images. The main difference from the original ZOC method is that we will provide prompts about unseen candidates of attributes instead of generating them using an image description generator. When we can use auxiliary samples, we train **Linear Probe** as in Radford et al. (2021). Also, we compared our method with **Red PANDA** (Cohen et al., 2022) which can learn to ignore specific attributes in the image.

**Metrics** We use three metrics to evaluate the performance of the methods: the Area Under Receiver Operating Characteristics (AUROC), the Area Under the Precision Recall Curve (AUPRC), and the False Positive Rate at the 95% true positive rate (FPR95). AUROC and FPR95 are commonly used for the anomaly detection or out-of-distribution detection task (Ming et al., 2022). And we also use AUPRC because some datasets are imbalanced, with a significant disparity between the number of normal and abnormal examples.

---

[1] https://github.com/openai/CLIP

Table 1: The anomaly detection performance on Colored MNIST dataset. We do not use additional data other than normal training set. For details, please refer to the main text.

| Method | Anom. Prompt | Number | | | Color | | |
|---|---|---|---|---|---|---|---|
| | | AUROC ↑ | AUPRC ↑ | FPR95 ↓ | AUROC ↑ | AUPRC ↑ | FPR95 ↓ |
| **No guidance** | | | | | | | |
| $k$NN | - | **0.880** | **0.879** | **0.617** | 0.817 | 0.878 | 0.442 |
| MSAD | - | 0.582 | 0.551 | 0.757 | **1.000** | **1.000** | **0.001** |
| **Guide** | | | | | | | |
| CLIP (MCM) | × | 0.549 | 0.499 | 0.877 | 0.892 | 0.942 | 0.460 |
| CLIP (ZOC) | ○ | 0.981 | 0.982 | 0.112 | **1.000** | **1.000** | **0.000** |
| | △ | 0.933 | 0.923 | 0.406 | **1.000** | **1.000** | **0.000** |
| LAFT AD (ours) | ○ | **0.989** | **0.990** | **0.066** | **1.000** | **1.000** | **0.000** |
| | △ | 0.984 | 0.985 | 0.089 | **1.000** | **1.000** | **0.000** |
| **Ignore** | | | | | | | |
| LAFT AD (ours) | ○ | **0.938** | **0.929** | **0.279** | 0.989 | 0.989 | 0.026 |
| | △ | 0.935 | 0.925 | 0.293 | **0.991** | **0.991** | **0.024** |

## 5.2 RESULTS ON COLORED MNIST

We used the colored version of the MNIST dataset (LeCun et al., 2010), similar to Arjovsky et al. (2019), to demonstrate our concept in the simplest way. We create a dataset that divides each digit of the MNIST and colors each split with red, green, and blue. In this way, the image of a colored MNIST consists of two attributes: number and color. We mark the numbers 0 to 4 as normal and the numbers 5 to 9 as abnormal. In addition, we label red as normal and green and blue as abnormal colors. In this setting, the training set consists of 0 to 4 and red images.

We assume that CLIP has learned visual concepts from a sufficiently large variety of image captions so that it can place images according to their degree of concept for a given concept axis. While MCM (Ming et al., 2022) assumes the text feature itself as "concept prototype", we use the "pairwise difference of concept prototypes" to find this axis. Figure 3 shows the brief overview of our desired transformation using the concept axis. If we choose one axis (number or color), we can just use $k$NN to detect anomaly with guidance to specific attributes.

Table 1 shows the main results on Colored MNIST. The table is divided into three groups: "no guidance", "guide", and "ignore". The "no guidance" group shows the performance of anomaly detection methods that do not provide any guidance and can be thought of as the default performance for that attribute. The "guide" group displays the performance of methods that can be directed to focus on a particular attribute, so it shows the performance when guided to the attribute corresponding to the label. The "ignore" group shows the performance of disregarding attributes other than the one to be evaluated, so it shows the performance of ignoring attributes that do not correspond to the label. For example, performance in the attributes "Number" means ignoring the attribute "Color" and vice versa. And the "Anomaly Prompt" column indicates the method uses the text prompt for anomalous attributes. The ○ means that the method uses exact anomaly prompts (in this case, 'green' and 'blue' for color attributes), and the △ means that the method also uses other candidate anomaly prompts (e.g. 'purple', 'orange', 'black', etc.). This format is used throughout the paper.

As can be seen from the table, the performance of the guidable methods is generally higher than the performance of the non-guidable methods, and our method has the best performance among them. It is notable that ZOC's performance drops significantly when given candiate anomaly prompts in addition to the exact anomaly prompts, while our method does not make much difference. This is a problem mentioned in Ming et al. (2022), where the performance of the methods that calculate anomaly scores based on image-text similarity in CLIP is highly affected by inaccurate prompts. In contrast, our method uses the prompt to calculate only the transformation of the image features(Equation 5), and the normality is actually determined through the images, so we can see that the performance is similar even with some inaccurate prompts. The important thing in image anomaly detection is to find anomalies that are different from normal images, so we can verify that our approach works effectively. We can also observe that when we use our method to guide one attribute, the other attributes are actually ignored, which we summarize further in Appendix B. In the "ignore" group of the table, it shows that ignoring one attribute increases the performance for the other attribute. This is because the Colored MNIST very clearly consists of only two attributes. In real-world datasets, the behavior is slightly different, which we discuss further in the following sections.

In summary, on the simple Colored MNIST dataset, we demonstrated that our method can leverage language to provide guidance on normality without additional training.

Table 2: The anomaly detection performance on Waterbirds dataset. We do not use additional data other than normal training set. For details, please refer to the main text.

| Method | Anom. Prompt | Bird | | | Background | | |
|---|---|---|---|---|---|---|---|
| | | AUROC ↑ | AUPRC ↑ | FPR95 ↓ | AUROC ↑ | AUPRC ↑ | FPR95 ↓ |
| **No guidance** | | | | | | | |
| kNN | - | **0.772** | **0.893** | **0.618** | 0.704 | 0.651 | 0.849 |
| MSAD | - | 0.615 | 0.275 | 0.833 | **0.855** | **0.826** | **0.504** |
| **Guide** | | | | | | | |
| CLIP (MCM) | × | 0.867 | 0.946 | 0.468 | 0.845 | 0.836 | 0.619 |
| CLIP (ZOC) | ○ | 0.927 | 0.971 | 0.276 | 0.961 | 0.963 | 0.231 |
| | △ | 0.920 | 0.966 | 0.363 | 0.951 | 0.952 | 0.315 |
| LAFT AD (ours) | ○ | **0.945** | **0.981** | **0.242** | **0.970** | **0.973** | **0.179** |
| | △ | 0.933 | 0.972 | 0.267 | 0.962 | 0.966 | 0.213 |

Table 3: The anomaly detection performance on Waterbirds dataset. We use a few or full additional data other than normal training set for training. For details, please refer to the main text.

| Method | # Shots / Subset | Bird | | | Background | | |
|---|---|---|---|---|---|---|---|
| | | AUROC ↑ | AUPRC ↑ | FPR95 ↓ | AUROC ↑ | AUPRC ↑ | FPR95 ↓ |
| **Guide** | | | | | | | |
| Linear Probe | Full | 0.756 | 0.555 | 0.714 | 0.969 | 0.973 | 0.136 |
| LAFT AD (ours) | 0 | 0.945 | 0.981 | 0.242 | 0.970 | 0.973 | 0.179 |
| + CoOp | 1 | 0.934 | 0.974 | 0.259 | 0.953 | 0.961 | 0.288 |
| | 4 | 0.943 | 0.982 | 0.203 | 0.976 | 0.976 | 0.126 |
| | 8 | 0.945 | 0.980 | 0.207 | 0.981 | 0.983 | 0.097 |
| | 16 | 0.947 | 0.980 | 0.201 | 0.983 | 0.984 | 0.084 |
| | 128 | **0.954** | **0.983** | **0.177** | **0.991** | **0.992** | **0.034** |
| **Ignore** | | | | | | | |
| Red PANDA | Full | 0.612 | 0.610 | 0.882 | 0.717 | 0.722 | 0.824 |
| LAFT AD (ours) | 0 | 0.773 | 0.889 | 0.559 | 0.693 | 0.856 | 0.735 |
| + CoOp | 1 | 0.852 | 0.941 | 0.505 | 0.673 | 0.609 | 0.789 |
| | 4 | 0.885 | 0.947 | 0.389 | 0.891 | 0.897 | 0.523 |
| | 8 | 0.903 | 0.958 | 0.326 | 0.932 | 0.935 | 0.344 |
| | 16 | 0.918 | 0.969 | 0.305 | 0.952 | 0.955 | 0.258 |
| | 128 | **0.932** | **0.976** | **0.252** | **0.971** | **0.974** | **0.167** |

## 5.3 RESULTS ON WATERBIRDS

The Waterbirds dataset (Sagawa et al., 2019) is commonly used in studies focused on spurious correlation and representation disentanglement. The dataset consists of two primary attributes: bird (waterbird / landbird) and background (water / land). Naturally, the training set has a very strong correlation between birds and backgrounds, whereas the test set has an equal ratio of birds to backgrounds. We specify waterbirds and water backgrounds as a normal training set. Table 2 summarizes the results for the dataset. The trends observed in the Colored MNIST experiment remain consistent in the results, demonstrating that our method is applicable to real-world datasets. The one difference is that ignoring one attribute does not directly improve the performance of other attributes, as shown in the ignore group of Table 3.

To improve performance, we employ the prompt learning technique Context Optimization (CoOp; Zhou et al., 2022), in order to accurately capture the concept difference without prompt bias. See Appendix A for the details of CoOp. To train the prompt, we randomly select a few auxiliary samples from each train subset. In the practical application, we are often able to acquire samples that are not in the training set, and we can benefit from this. The results are summarized in Table 3. To use as a baseline, we trained Linear Probe and Red PANDA using all the data in the train set.

Based on our findings, it is clear that using the image features directly from CLIP and training a linear classifier does not outperform our model. This demonstrates that properly transforming the features is more effective for the desired downstream task. Furthermore, we observed that with only four samples for each subset, we are able to effectively learn the prompt to further improve the performance. These findings are consistent with those reported in (Zhou et al., 2022), thus highlighting the effectiveness of combining CoOp with our method. Additionally, as addressed in (Cohen et al., 2022), fine-tuning of the feature extractor is limited in real-world datasets.

Table 4: The anomaly detection performance on CelebA dataset. We do not use additional data other than normal training set. For details, please refer to the main text.

| Method | Anom. | Blond | | | Eyeglasses | | | Young | | |
|---|---|---|---|---|---|---|---|---|---|---|
| | Prompt | AUROC ↑ | AUPRC ↑ | FPR95 ↓ | AUROC ↑ | AUPRC ↑ | FPR95 ↓ | AUROC ↑ | AUPRC ↑ | FPR95 ↓ |
| **No guidance** | | | | | | | | | | |
| kNN | - | **0.865** | **0.974** | **0.541** | **0.778** | 0.185 | 0.677 | **0.701** | **0.477** | **0.863** |
| MSAD | - | 0.827 | 0.964 | 0.637 | 0.742 | **0.969** | **0.659** | 0.528 | 0.291 | 0.974 |
| **Guide** | | | | | | | | | | |
| CLIP (MCM) | × | 0.848 | 0.972 | 0.709 | 0.323 | 0.044 | 0.989 | 0.460 | 0.234 | 0.967 |
| CLIP (ZOC) | △ | 0.908 | 0.980 | 0.642 | **0.989** | **0.963** | **0.003** | 0.760 | 0.592 | **0.713** |
| LAFT AD (ours) | △ | **0.930** | **0.987** | **0.351** | **0.989** | 0.923 | 0.038 | **0.798** | **0.634** | 0.748 |

Table 5: The anomaly detection performance on CelebA dataset. We transform the image features using LAFT to ignore the Male attribute. Then, we use the transformed features for the ZOC anomaly detection. For details, please refer to the main text.

| Method | Blond | | | Eyeglasses | | | Male | | |
|---|---|---|---|---|---|---|---|---|---|
| | AUROC ↑ | AUPRC ↑ | FPR95 ↓ | AUROC ↑ | AUPRC ↑ | FPR95 ↓ | AUROC ↑ | AUPRC ↑ | FPR95 ↓ |
| CLIP (ZOC) | 0.908 | 0.980 | 0.642 | 0.989 | 0.963 | 0.003 | 0.996 | 0.997 | 0.010 |
| + LAFT | 0.916 | 0.982 | 0.531 | 0.989 | 0.957 | 0.008 | **0.508** | **0.618** | **0.996** |

## 5.4 RESULTS ON CELEBA

To verify that our method works in multi-attribute settings, we use the CelebA dataset (Liu et al., 2015), which contains over 200K celebrity images with 40 attribute labels. For the normal training set, we select three attributes: Blonde_Hair, (No) Eyeglasses, and Young. The results are displayed in Table 4. While the tendencies of Blond_Hair and Young are similar to previous experiments, the results of Eyeglasses are slightly different. This is because CLIP can classify almost perfectly whether a person is wearing glasses or not. Therefore, using images to define normality provides disturbing information for the attribute. And, notably, the performance for the Young attribute is not good for all models. Similar results are also reported in Gannamaneni et al. (2023). This suggests that CLIP may have difficulty conceptualizing age, a limitation that also affects our method, which relies on CLIP's image-text alignment.

## 6 LIMITATIONS AND DISCUSSION

**Ignore attributes using LAFT**    Unlike in a simple Colored MNIST dataset, we observe that ignoring one attribute using LAFT without CoOp does not improve the anomaly detection performance of the other attribute in real-world datasets. However, as seen in Appendix B, the LAFT actually suppresses the attribute to be ignored. We hypothesize this phenomenon because it is difficult to remove all attribute-related information in the embedding space using only text prompts. On the other hand, guiding the attribute is relatively easy, because LAFT only needs to capture the primary information about the attribute. We found that performance improved when we used a genetic algorithm to select the appropriate pair from the given prompt pairs. Selecting the appropriate prompt pair would be our future work.

**Using LAFT with other methods**    Our proposed LAFT method can be used as a feature transformation module in other tasks or methods. Basically, we expect that it can be applied to any vision model that requires a feature extractor. As a simple proof of concept, we apply the LAFT method to ZOC for anomaly detection. The results in Table 5 show that we can suppress Male attribute from the image features without significant impact on other attributes. Applying LAFT to other downstream tasks would be an interesting future work.

## 7 CONCLUSION

In this paper, we propose the novel feature transformation method LAFT to adapt pre-trained CLIP image features for the target task. Our LAFT AD approach demonstrates how language can guide normality detection by combining LAFT with $k$NN for anomaly detection. We show that defining normality through image features is crucial for image anomaly detection and outperforms language-based methods across various datasets.

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

## A  EXPERIMENTAL DETAILS

**Candidate attribute values**  We use the actual class names from the dataset, if available. For example, for the number attribute in the Colored MNIST dataset, we can give exactly '0' to '9' and 'zero' to 'nine'. And for the Waterbirds dataset, we use 200 bird class names provided by the dataset. However, for attributes that cannot get exact or few labels (e.g., blonde hair), we get candidate labels from the Internet.

**Details of CoOp**  We use context length $M = 16$ for learning the CoOp prompt. To train the prompt, we train $3000$ steps with the Adam optimizer at a learning rate of $1e-3$ and do not use the learning rate scheduler.

## B  ADDITIONAL EXPERIMENTS

Table 6: The anomaly detection performance of LAFT AD on Colored MNIST dataset. We do not use any additional data other than the normal training set. Bold text indicates that the performance should be high, and underlined text indicates that the performance should be low.

| Attribute | Anom. Prompt | Number AUROC ↑ | AUPRC ↑ | FPR95 ↓ | Color AUROC ↑ | AUPRC ↑ | FPR95 ↓ |
|---|---|---|---|---|---|---|---|
| **No guidance** | | | | | | | |
| *k*NN | - | 0.880 | 0.879 | 0.617 | 0.817 | 0.878 | 0.442 |
| **Guide** | | | | | | | |
| Number | ○ | **0.989** | **0.990** | **0.066** | 0.543 | 0.644 | 0.735 |
| | △ | **0.984** | **0.985** | **0.089** | 0.553 | 0.654 | 0.721 |
| Color | ○ | 0.515 | 0.506 | 0.878 | **1.000** | **1.000** | **0.000** |
| | △ | 0.519 | 0.513 | 0.770 | **1.000** | **1.000** | **0.000** |
| **Ignore** | | | | | | | |
| Number | ○ | 0.619 | 0.600 | 0.743 | **0.989** | **0.989** | **0.026** |
| | △ | 0.586 | 0.562 | 0.747 | **0.991** | **0.991** | **0.024** |
| Color | ○ | **0.938** | **0.929** | **0.279** | 0.654 | 0.738 | 0.596 |
| | △ | **0.935** | **0.925** | **0.293** | 0.698 | 0.782 | 0.565 |

Table 7: The anomaly detection performance of LAFT AD on Waterbirds dataset. We do not use any additional data other than the normal training set. Bold text indicates that the performance should be high, and underlined text indicates that the performance should be low.

| Attribute | Anom. Prompt | Bird AUROC ↑ | AUPRC ↑ | FPR95 ↓ | Background AUROC ↑ | AUPRC ↑ | FPR95 ↓ |
|---|---|---|---|---|---|---|---|
| **No guidance** | | | | | | | |
| *k*NN | - | 0.772 | 0.893 | 0.618 | 0.704 | 0.651 | 0.849 |
| **Guide** | | | | | | | |
| Bird | ○ | **0.945** | **0.981** | **0.242** | 0.662 | 0.655 | 0.855 |
| | △ | **0.929** | **0.975** | **0.303** | 0.661 | 0.650 | 0.847 |
| Background | ○ | 0.641 | 0.857 | 0.856 | **0.970** | **0.973** | **0.179** |
| | △ | 0.658 | 0.866 | 0.854 | **0.962** | **0.966** | **0.213** |
| **Ignore** | | | | | | | |
| Bird | ○ | 0.678 | 0.615 | 0.796 | **0.693** | **0.856** | **0.735** |
| | △ | 0.675 | 0.612 | 0.800 | **0.693** | **0.856** | **0.737** |
| Background | ○ | **0.773** | **0.889** | **0.559** | 0.604 | 0.562 | 0.914 |
| | △ | **0.769** | **0.888** | **0.576** | 0.605 | 0.561 | 0.904 |

The results in Table 6 and Table 7 show that when we guide one attribute, the other is implicitly ignored, and vice versa.

# C  ALGORITHM

```python
# model: the CLIP model
# prompts: the list of prompts provided by the user
# train_images: the collection of normal images
# test_images: the collection of images to be tested
# d: the number of principle axis

# Compute attribute subspace
text_features = model.encode_text(prompts)
pair_diffs = pairwise_difference(text_features)
basis = pca(pair_diffs, d)

# Encode images
train_features = model.encode_image(train_images)
test_features = model.encode_image(test_images)

# Guide
train_laft_features = inner_projection(train_features, basis)
test_laft_features = inner_projection(test_features, basis)

anomaly_scores = knn(train_laft_features, test_laft_features)

# Ignore
train_laft_features = orthogonal_projection(train_features, basis)
test_laft_features = orthogonal_projection(test_features, basis)

anomaly_scores = knn(train_laft_features, test_laft_features)
```

