# OpenReview forum: "Guide Your Anomaly with Language"
_ICLR.cc/2024/Conference — ICLR 2024 Conference Withdrawn Submission_

### Official Review · Reviewer_e9uy · 2023-10-14

**Soundness:** 2 fair
**Presentation:** 1 poor
**Contribution:** 2 fair
**Rating:** 3
**Confidence:** 5

**Summary:**

The paper introduces an approach to anomaly detection that leverages natural language guidance. By incorporating user priors and harnessing CLIP feature space, the authors incorporate guidance into data representations. The authors demonstrate the efficacy of their approach through an evaluation process, showcasing advancements over existing anomaly detection methods.

**Strengths:**

The paper tackles a pivotal challenge: effectively handling nuisance factors of variation while integrating user priors. By delving into the complexities of nuisance factors, the authors confront a critical aspect often overlooked in previous research. Additionally, the incorporation of user priors adds a crucial layer of customization, allowing for a more nuanced and contextually sensitive understanding of normality, which is vital for real-world applications.

**Weaknesses:**

The paper has major weaknesses that should be addressed.
1. The manuscript suffers from a significant writing issues that impacts its overall clarity and persuasiveness. The importance of the addressed problem is not sufficiently emphasized in the introduction. Rather than succinctly presenting the proposed methodology, the introduction reads more like an extended related work section. As a result, the paper fails to effectively communicate the criticality of the nuisance factor of variation and the integration of user priors in the context of anomaly detection.

2. The paper lacks clear guidance on obtaining robust user priors, a crucial aspect for successful implementation. While the authors highlight the incorporation of user priors as a key strength, they do not offer sufficient guidance or methodologies for acquiring these strong priors. A more detailed discussion on how to obtain and integrate such priors is needed to enhance the practical applicability of the proposed approach.

3. The experimental section of the paper could benefit from a more comprehensive and insightful analysis of the proposed method's capabilities. For instance, the authors do not adequately explore the impact of different prompts, a factor that can significantly influence the effectiveness of the approach. The "triangle setting", where the authors claim to have employed alternative anomaly prompts, lacks in-depth presentation and explanation. Additionally, the paper does not conduct an ablation study on the number of principal components used in the experiments. An implementation details section could provide invaluable insights for readers seeking to replicate and extend the presented work. Such details are crucial for ensuring reproducibility and promoting a deeper understanding of the method's logic.

Currently, the manuscript is far from the acceptance threshold and should be greatly revised before resubmitting.

**Questions:**

See weaknesses.

---

### Official Review · Reviewer_h8da · 2023-10-16

**Soundness:** 3 good
**Presentation:** 3 good
**Contribution:** 2 fair
**Rating:** 5
**Confidence:** 3

**Summary:**

This paper uses CLIP to perform anomaly detection in images, using textual description embeddings to transform the feature space to emphasise or ignore certain attributes followed by anomaly scoring with a knn estimator.

**Strengths:**

Type control for anomaly detection is an interesting problem with important applications.

The methodology is well explained and easy to understand, and the use of text prompts to transform the feature space, instead of for similarity matching, is novel.

**Weaknesses:**

An inherent flaw of the methodology is that it implicitly assumes knowledge of the presentation of anomalies at test time. For example, choosing to guide using the eyeglasses attribute in CelebA assumes that there cannot be an anomaly that follows the normal patterns in the eyeglasses attribute, which is not always a realistic assumption.

Another flaw is that anomalies often manifest as small aberrations from the normal patterns, often in only a small region of an image, which may not be easily defined by one clear attribute. The attributes used in these experiments are mostly very large-scale, global features such as hair colour or background, which means its difficult to judge how this method could perform in more realistic AD problems.

Comparison with baseline methods is also quite lacking. The authors identify several AD with type control methods in the literature review but only considers RedPanda among them in the experiments, and that is only in one experiment setting.

**Questions:**

How does the relevance of text prompts affect performance? The authors claim that using density-based scoring means that the text prompts do not need to be entirely accurate, but have no experiments to support this.


It is not entirely clear why dimensionality reduction with PCA achieves the stated goal of removing irrelevant information about the target attribute. What is the intuition behind this?

What is the reason for the large discrepancy between the number of components to use for guiding vs. ignoring an attribute? Does this hyper-parameter have a significant effect on performance?

---

### Official Review · Reviewer_AWhP · 2023-10-25

**Soundness:** 1 poor
**Presentation:** 2 fair
**Contribution:** 1 poor
**Rating:** 3
**Confidence:** 4

**Summary:**

The manuscript deals with anomaly detection atop langue-vision models. The proposed method increases sensitivity to specific anomaly types for which prior knowledge is available.
This is done by encoding the input image into a feature vector which is then projected into a subspace where expected anomalies are well separated from the normal data.
 The required projection is based on of principal components of the encoded textual templates.
Anomalies are then detected according to non-parametric density estimation implemented by the k-NN algorithm. The proposed method achieves competitive results when encountered anomalies are aligned with expected anomalies described within the textual templates.

**Strengths:**

S1. Anomaly detection is an important issue.

S2. The proposed method achieves competitive results when test anomalies correspond to prior knowledge about anomalies.

**Weaknesses:**

W1. The described method essentially identifies previously unseen classes for which prior knowledge is available (such as textual descriptions). This is more similar to zero-shot learning than to anomaly detection. Therefore, this manuscript is wrongly framed.

W2. The method assumes complete knowledge about anomalies that may appear during inference (as described in Sec 4.1). However, this is an unreasonable assumption in many practical applications which limits the manuscript contribution.

W3. The presented experiments do not consider cases where anomalies do not follow expected deviation patterns. Presenting evidence for such generalization capabilities would be beneficial.
E.g. if normal samples are human faces with specific hair type and prior knowledge expects anomalies will have different hair colors, can the model detect CIFAR10 samples as anomalies?

W4. Missing ablation studies for method hyperparameters such as the number of principal components used.

W5. Section 3 appears detached from the rest of the manuscript. For example, the mutual information mentioned in Eq 2 is hardly relatable with the method described in Sec 4. The connection should be emphasised more clearly.

**Questions:**

Please see the weaknesses.

---

### Official Review · Reviewer_rVqL · 2023-10-28

**Soundness:** 2 fair
**Presentation:** 2 fair
**Contribution:** 2 fair
**Rating:** 3
**Confidence:** 5

**Summary:**

The paper proposes an anomaly detection method using text to support the definition of normality via CLIP-like vision-language models. The proposed method requires proper text descriptions of the normality concept, e.g., “a photo of a person with brown hair”. The text descriptions are embedded via the pretrained CLIp text encoder and then are used to guide the projection of the image features. The transformed image feature space is expected to be composed of anomaly detection-relevant features. For scoring anomalies, they use a KNN-based model to compare the test samples to training samples in the projected feature space. The proposed method is evaluated on three image datasets, i.e., MNIST, Waterbirds, and CelebA, to showcase its effectiveness.

**Strengths:**

1. The paper combines text inputs and images in anomaly detection. By utilizing both text and image as inputs, the proposed method can understand the normality concept better than methods relying either on text or images.

2. The proposed method achieves remarkable results in the designed experiments.

**Weaknesses:**

1. The major contribution of the proposed method is combining the text inputs and images for anomaly detection. However, on the one hand,  the proposed method requires training images and loses the zero-shot ability compared with other CLIP-based zero-shot anomaly detection methods. On the other hand, the proper text descriptions are not always available in many applications, e.g., detecting the defects in a part in manufacturing.

2. The presentation can be improved. From the presentation, I see how the method works. However, it is not clearly claimed why the proposed method is more advantageous than other methods. The motivation behind the design choices is vague.

3. In the empirical evaluation, the authors simulate the experiments on three datasets with detailed attribute labels. It would be more convincing to see the evaluation on some popular benchmark datasets, e.g., MVTEC, and CIFAR100. How would you design the text prompts in datasets where the detailed attribute labels are not available?

4. The proposed method uses text to guide the transformation of image features. Related works about learning the transformations are missing, e.g. [1,2]. Related works about transformation-based anomaly detection should be discussed, e.g., [3,4].

[1] Alex Tamkin, Mike Wu, and Noah Goodman. Viewmaker networks: Learning views for unsupervised representation learning. In ICLR, 2021.

[2] Chen Qiu, Timo Pfrommer, Marius Kloft, Stephan Mandt, and Maja Rudolph. Neural transformation learning for deep anomaly detection beyond images. In International Conference on Machine Learning, 2021

[3] Golan, I. and El-Yaniv, R. Deep anomaly detection using geometric transformations. In Advances in Neural Information Processing Systems, 2018.

[4] Wang, S., Zeng, Y., Liu, X., Zhu, E., Yin, J., Xu, C., and Kloft, M. Effective end-to-end unsupervised outlier detection via inlier priority of discriminative network. In Advances in Neural Information Processing Systems, 2019.

[5] Bergman, L. and Hoshen, Y. Classification-based anomaly detection for general data. In International Conference on Learning Representations, 2020.

**Questions:**

Please see the weaknesses above.